# Network Medicine Approach for Analysis of Alzheimer’s Disease Gene Expression Data

**DOI:** 10.3390/ijms21010332

**Published:** 2020-01-03

**Authors:** David Cohen, Alexander Pilozzi, Xudong Huang

**Affiliations:** Neurochemistry Laboratory, Department of Psychiatry, Massachusetts General Hospital and Harvard Medical School, Charlestown, MA 02129, USA; dscohen@protonmail.com (D.C.); APILOZZI@mgh.harvard.edu (A.P.)

**Keywords:** Alzheimer’s disease, network medicine, gene expression, neurodegeneration, neuroinflammation

## Abstract

Alzheimer’s disease (AD) is the most widespread diagnosed cause of dementia in the elderly. It is a progressive neurodegenerative disease that causes memory loss as well as other detrimental symptoms that are ultimately fatal. Due to the urgent nature of this disease, and the current lack of success in treatment and prevention, it is vital that different methods and approaches are applied to its study in order to better understand its underlying mechanisms. To this end, we have conducted network-based gene co-expression analysis on data from the Alzheimer’s Disease Neuroimaging Initiative (ADNI) database. By processing and filtering gene expression data taken from the blood samples of subjects with varying disease states and constructing networks based on that data to evaluate gene relationships, we have been able to learn about gene expression correlated with the disease, and we have identified several areas of potential research interest.

## 1. Introduction

Alzheimer’s disease (AD) is the most widespread diagnosed cause of dementia in the elderly [1]. It is a progressive neurodegenerative disease that causes memory loss as well as other detrimental symptoms and is always fatal. Due to increased lifespans across the globe, this already common disease is expected to become drastically more prevalent in the near future unless intervention occurs. In the United States alone, projections show the prevalence in individuals aged 65 years or older increasing from 4.7 million in 2010 to 13.8 million in 2050 [2]. At the present time, a wide variety of research is being conducted in order to counteract the growing AD epidemic; despite this, there is currently no known effective method of treatment or prevention. Researchers are exploring this disease from many different perspectives in order to better understand its underlying mechanisms responsible for it [3]. In this paper, the application of network medicine (NM) on AD will be explored. NM is a constantly developing field of research that strives to connect the various genetic, molecular and environmental drivers of diseases such as AD to as many involved components as possible. Ideally, having a more complete picture of disease-related pathways will grant more avenues for potential treatment than reducing the problem to single components or genes [4]. A significant focus of network medicine involves inspecting the complex interactions between genes underlying diseases, such as AD [5,6,7]. Diseases occur at varying biological complexities and therefor it is crucial to understand the networks of genes underlying a disease. The more holistic approach of NM advances drug targeting by eliminating the reliance on single components or genes, and instead allows for targeting networks of interacting components or genes. It also allows for the analysis of potential off target effects, which is very useful for drug development. NM can provide very insightful results on datasets such as gene expression data [8,9]. While studies involving gene expression data from AD patients are not rare, there are comparatively few that conduct network-based analysis on that data. Much of the work involving Alzheimer’s related network analysis apply the techniques to protein interactomes [10,11] or the interactomes of existing drugs/drug-targets [12]. Of some involved in co-expression analysis, Mostafavi et al. constructed networks based on expression data obtained from the dorsolateral prefrontal cortex of nonaffected individuals and those afflicted with mild cognitive impairment (MCI) or AD; genes were clustered into, and analyzed as, modules [13]. Here, we examine co-expression based on blood samples, looking at transcript-level interactions found in a heavily filtered subset of the expression data.

## 2. Results

Figure 1A is the visual representation of the DyNet network associated with positive gene expression correlations. Figure 1B is the visual representation of the DyNet network associated with negative gene expression correlations. Figure 2A is the visual representation of the Diffany network associated with positive gene expression correlations. Figure 2B is the visual representation of the Diffany network associated with negative gene expression correlations. Figure 3 and Figure 4 are the simple correlation networks for the different disease states ordered by correlation type. All networks are included as Appendix A. All numbers correspond to genes in Table 1 by key. The 49,293 transcripts from the original expression dataset were filtered in two stages, using ANOVA (groups were NC, MCI, AD, *p* < 0.1) and additional thresholds for expression level. The final number of transcripts passing filtration was 50. These 50 genes are described in Table 1. Transcript names that are repeated in the dataset represent transcripts that are associated with multiple. The probe set ID for each transcript is listed below its name. Differences in expression levels relative to NC are listed, with *p* values for the differences that are greater than 0.1 shown below the overall direction of relative expression. A “-” indicates the *p* value exceeded 0.75, regardless of direction (Welch *t*-test, FDR adjusted *p* value).

Looking at Table 1, we can see that in many of the genes that are significantly upregulated and downregulated in AD individuals are regulated in the same direction in MCI individuals, though there are some notable exceptions such as haptoglobin, along with genes whose expression changes are not necessarily significant between the NC and MCI states.

From the correlation matrices generated from the expression data of the genes in Table 1, DyNet networks were generated. These networks indicate nodes whose connections change between states (i.e., connections in one network that are not present in the other network or network). The deep red nodes, such as the haptoglobin transcript node #38, experience a greater degree of connectivity disruption (i.e., connections added or removed) between the different disease states. The light red/white nodes experience more consistent connection (i.e., few connections added or removed) between the different disease states. Edges are present in the DyNet networks where any network (NC, MCI, or AD) has an edge.

Table 2 contains the top rewiring scores for the DyNet network associated with positive gene expression correlations. Table 3 contains the top rewiring scores for the DyNet network associated with negative gene expression correlations. Both tables were cut off at the score of 5. Some genes with scores lower than five were considered for further analysis; those with scores nearing five were taken into consideration, one such gene being OSBP2.

To determine the differences in co-expression relative to NC individuals that are common to both MCI and AD participants, Diffany networks were constructed. A red edge indicates that an edge present in the NC network is absent in both the MCI and AD networks. A green edge indicates that an edge absent in the NC network is present in both the MCI and AD networks. From the positive co-expression Diffany network (Figure 2A), we can find that certain transcripts such as folate receptor 3 (#23) gain positive co-expression-relationships to a host of different transcripts in MCI and AD participants whereas one MHC transcript (#31) loses positive co-expression relationships. Many other transcripts exhibit combinations of gained and lost relationships.

The negative co-expression Diffany network (Figure 2B) follows the same principles as the positive co-expression network; green edges indicate a negative co-expression relationship present in only the MCI and AD networks, while red edges indicate a negative co-expression relationship absent in only the MCI and AD networks. From this network, we find that transcripts such as *ANKRD22* (#11) are no longer negative co-expressed with several other transcripts such as the three *DDX3Y* (#15, #29, #30) and one *KDM5D* (#7) transcript in the MCI and AD states. While there are no transcripts that purely gain negative co-expression relationships, transcripts such as the two V2-13 protein fragments commonly gain negative co-expression relationships with the three haptoglobin transcripts (#14, # 25, #38) and lose negative co-expression relationships with *GATA2* (#3).

For the purpose of examining the general co-expression relationships between genes, three additional co-expression networks for created with a correlation threshold of 0.3 to eliminate the weakest correlations; transcripts (nodes) with correlation coefficients above 0.3 in the positive correlation networks and below −0.3 in the negative correlation networks have an edge between them. Red edges indicate higher absolute-value Pearson coefficients of correlation than more yellow edges (i.e., 0.9 and −0.9 will appear more deeply red than 0.4 and −0.4, respectively).

## 3. Discussion

### 3.1. Network Medicine Applied to Gene Expression Data

Through the mapping out of these genes in networks, we can better understand their relations to MCI and AD. The effectiveness of the network comparison tools utilized is supported by our results. Many of the genes that are duplicates/of the same family have similar rewiring scores; all of the instances of haptoglobin (*HP*), for example, have rewiring scores between 7 and 8.33 relative to the positive-correlation networks. These same groupings of genes are not connected in the Diffany networks, indicating that the relationships between them, which should be very strong, are not disrupted between disease states. Furthermore, while they themselves are not connected, gene families/groups have many other common gene connections, indicating they experience many of the same correlational changes between disease states.

It is demonstrated that an effective method was followed for gene selection due to the nature of some of the genes observed; many of the genes found to be significant have well-studied mechanisms for contributing to AD, and many others have significant regulatory function or are commonly expressed in the brain.

### 3.2. Highly Disrupted Genes

Of the overexpressed genes, haptoglobin, which had the highest rewiring score between the positive correlation networks, has been shown to be affected by AD in previous research [14]. Indeed, serum levels of HP are observed in significantly higher quantities in individuals with AD as well as MCI [15]. It is known to be partially responsible for suppressing certain types of inflammatory responses [16]. Inflammation, both systemic and localized to the central nervous system, is widely accepted to be a contributor to Alzheimer’s disease [17]. The fact that it is only expressed in AD patients may be due to it is released in response to AD-specific inflammation.

Folate receptor 3 (gamma) is another gene of interest that exhibited a high degree of correlational disruption. It encodes a member of the folate receptor family of proteins, which have a high affinity for folic acid and folate intake has been correlated with reduced risk of AD [18]. Folate deficiency contributes to hyperhomocysteinemia, which is a risk factor for Alzheimer’s as well as other neurological disorders [19].

*OSBP2* is a gene whose product binds to Oxysterol, which is known for its contribution to cholesterol disequilibrium. High cholesterol is a known risk factor for Alzheimer’s disease, but cholesterols themselves cannot penetrate the blood brain barrier, making the mechanism by which hypercholesterolemia contributes to the disease somewhat obscure. On the other hand, oxysterols, which are oxidized cholesterol metabolites, are able to enter the brain [20].

Cystathionine beta synthase (CBS) has been shown to be associated with AD due to its role in homocysteine metabolism. It, in conjunction with two other enzymes, is responsible for the metabolism of homocysteine, with accumulation of homocysteine, hyperhomocysteinemia, being a known risk factor for AD [21]. CBS is notably overexpressed in individuals with Alzheimer’s disease, indicating the possibility that some other portion of the cysteine metabolic pathway is disrupted, and an increase in CBS is a response. Additionally, the metabolic activity of CBS increases levels of H_2_S, which is known to be neuroprotective, in the brain [22].

*TBC1D22B* is a gene with a notable TBC domain; within the networks it experienced a fairly high degree of connection disruption based on DyNet. TBC domain proteins are primarily GTPase activating proteins for the small GTPase Rab, and defective TBC proteins are implicated in a variety of human diseases. As GTPase activity is a regulator of other cellular functions, those genes regulated by *TBC1DD2B* are of interest as potential contributors to AD and MCI [23].

For negatively expressed genes, GATA binding protein 2 is known to be an essential transcription factor for neuroglobin; *GATA-2* knockdown causes significant drops in neuroglobin expression. Neuroglobin has been observed to have a protective effect on neural cells and has been implicated in reducing the severity of AD [24]. It experiences the most rewiring relative to the negative correlation networks. Tropomodulin, a regulator of actin, has been found to be important to the proper development of neural dendrites [25]. As a final example shisa member family 4 is more esoteric than some of the other genes but is shown to be highly expressed in the brain [26].

### 3.3. Notable Connections and Clusters

#### 3.3.1. Y-Linked Regulators

While those genes highlighted by the DyNet and Diffany networks are certainly significant and show some unique patterns of correlation disruption between NC, MCI, and AD conditions, the primary utility of network mapping is to highlight strong relationships between genes. Notably, some genes were excluded entirely in both the positive and negative correlation Diffany networks. These genes have many connections within the base networks but are notably genes associated with basic cellular function. These include: *EIF1AY*, *DDX3Y*, *USP9Y*, and *KDM5D*. *EIF1AY* is a gene located on the Y chromosome, which encodes a translation initiation factor thought to stabilize the binding of initiation Met-tRNA to the ribosome [27]. *DDX3Y* is a gene located on the Y chromosome that encodes a member of the DEAD-box RNA helicase family that is active in male germ cells [28]. The third member of this group of Y-linked genes is *USP9Y*, which is a protease which cleaves ubiquitin from ubiquitinylated proteins and ubiquitin-fused precursors [29]. The last of the Y-linked genes in this set is *KD5MD*, which is a male lysine-specific histone demethylase that regulates transcription factors that modulate the cell cycle [30]. As one might expect of these four genes, they are all directly connected on the positive-correlation networks of every disease condition, and all three are over-expressed in both MCI and AD individuals.

Looking at all three networks of positively correlated genes, there is a notable network of over 13 genes which are all related to each other. This group is mostly comprised of duplicates/variants of the Y-linked genes mentioned above, along with an additional Y-linked gene, *RPS4Y1*, which codes for a ribosomal protein [31]. All of these genes are in some fashion, be it direct-involvement or regulation, related to the gene expression process, are Y-linked and are themselves overexpressed in individuals with Alzheimer’s disease. The other items in this cluster are largely unidentified transcripts; perhaps their connection to the other Y-linked genes may aid in their identification.

Interestingly, in NC individuals the levels of Prostaglandin D2 synthase (*PTGDS*) expression are not significantly correlated to any gene in this study. However, in both the MCI and AD conditions, it is correlated to the entire cluster of Y-linked genes, albeit at only a moderate level. Prostaglandin is notable for being active in nervous system development and regeneration processes [32].

#### 3.3.2. Immune-System Involved

While the previously mentioned cluster is present in all three disease states, it is more well-defined in MCI and AD patients. In NC individuals, the cluster mixes with another that is normally mediated by the *BPI* (bactericidal/permeability-increasing protein) gene, which is a hub between two clusters in the MCI and AD condition networks. Notably, this hub contains the aforementioned BPI gene, as well has the haptoglobin genes (*HP*) and the *DEFA4* gene. *BPI* and *DEFA4* both code for components of the immune system; the latter is a defensin, which are known for disrupting microbial membranes, while the former is a protein involved in enhancing immune-cell bacterial recognition [33,34]. There may in fact be a connection between these immune genes and the haptoglobin gene, as findings suggest that haptoglobin has a role in the regulation of the immune system; haptoglobin deficient mice have a reduction in the presence of T and B cells, and they exhibit overall inhibited adaptive immune responses [35]. The immune system is known to have an impact on the pathogenesis of Alzheimer’s disease, making the aforementioned immune components, alongside other related pathways and regulators, a worthy target for future investigation [36]. It is unclear whether defensin’s connection to the greater translation cluster is a coincidence, or if their pathways are linked in some way; it is also unclear why this linkage seems to be more prevalent in those afflicted with any degree of cognitive impairment than cognitively normal individuals.

A somewhat related cluster involved in all three disease states contains the four transcripts associated with some portion of the Major Histocompatibility Complex (MHC). The MHC is responsible for presenting antigens along the cell surface for T-cell recognition [37]. While the relationship between the four is preserved throughout the three disease states, it breaks down somewhat in AD individuals, with the gene only identified to be somewhat related to the MHC (44% BLAST hit) becoming less correlated with the others. Also, it should be mentioned that overexpression of these MHC transcripts is far less significant in AD patients than in MCI patients, with substantial differences between the alpha and beta transcripts. Regardless, it further reinforces the idea that the immune system has an important role to play in the progression of cognitive decline, as the various immune-system related genes in the set studied are all overexpressed in AD patients.

#### 3.3.3. Under Expressed Regulatory Elements

Looking to the networks of negative correlations, there appear to be fewer hubs of interest than in the positive correlation networks. Central hubs of the networks for all three disease states include three instances of X-inactive specific transcript (*XIST*), which is downregulated in both MCI and AD individuals. XIST is a regulatory long non-coding RNA (lncRNA); XIST notably is involved in X-inactivation in females, possibly explaining why there are fairly consistent negative correlations between it and the Y-linked genes [38]. LncRNAs in general have been implicated in the pathogenesis of Alzheimer’s disease that are both up and down regulated in the disease [39]. Another hub that is present in all three is a *JARID1C* splice variant. JARD1C is integral to heterochromatin formation and replication. As heterochromatin cannot be transcribed, inhibiting its formation through the downregulation of *JARID1C* may be a potential mechanism for the overexpression of the host of genes connected to it [40]. Both XIST and JARD1C are known regulators of neural development and have been implicated in other neural/intellectual disorders [41]. Notably, the three *XIST* genes and the *JARD1C* gene are connected to most of the same genes in all three negative correlation networks; additionally, these correlations seem to weaken in AD subjects specifically, indicating potential interference by a competing regulatory pathway. Additionally, all four are themselves a separate non-connected subgraph on all three positive correlation networks, indicating that the expression of *XIST* and *JARD1C* are linked in some way, and whatever mutually regulates their expression may be a target of interest for future research.

#### 3.3.4. Folate Receptors

Aside from this central cluster of four genes in the negative-correlation networks, there is a pattern of interest involving the two folate receptor genes. While these genes are only present on the network of NC individuals based on relatively weak correlations to *DEFA4*, it becomes far more connected from MCI to AD individuals; they are connected to the three genes for *HP* as well as Defensin in MCI individuals with moderate correlations, and they are connected to over 17 different genes in AD patients with relatively strong correlations. Why the folate receptor is so central to the negative-correlation network in AD patients is unclear, though folate does have cited benefits against AD [18].

#### 3.3.5. Unidentified Transcripts

Another cluster of interest involves the majority of the unidentified/partially identified genes/gene products. In all three networks, these genes fully connected to each other and to both instances of the folate receptor gene. This could prove useful in further identification of the genes, and they may prove to be relevant to Alzheimer’s research given folate’s protective benefits against the disease [18].

## 4. Materials and Methods

### 4.1. Alzheimer’s Disease Neuroimaging Initiative

Data used in this paper originates from the Alzheimer’s Disease Neuroimaging Initiative (ADNI) database (adni.loni.usc.edu). The ADNI was launched in 2003 as a public–private partnership, led by Principal Investigator Michael W. Weiner, MD. An overarching goal has been for the ADNI to provide a collection of multi-categorical data that researches can utilize in order to better understand mild cognitive impairment (MCI) and AD. The ADNI has recruited over 800 adults, aged 55 to 90 from sites spanning both the US and Canada. In the initial stage of the study, approximately 200 cognitively normal individuals were followed for three years, 400 subjects with MCI were followed for three years, and 200 patients with early AD were followed for two years [42]. The project has gone on to involve further studies, adding in additional subjects and continuing the observation of previous participants [43]. ADNI data spans different categories including clinical data, MR image data, PET image data, genetics data, and biospecimen data. For the purpose of creating the network, gene expression data extracted from the participants’ blood samples will be the primary form of data used in this paper. Gene expression profiles were taken from 811 ADNI participants using the Affymetrix Human Genome U219 Array. Sixty-four samples were removed from the dataset as they did not pass quality control checks [44].

### 4.2. Network Medicine Applied to AD Gene Expression Data

For the network medicine analysis, microarray gene expression data was utilized in tandem with a diagnosis dataset. In order to properly utilize these data, diagnosis status (NC, MCI, or AD) was merged with the expression dataset. The diagnosis given on or nearest to the date of gene expression collection was assigned to each participant. The data was also cleaned prior to use. The gene expression dataset contains 49,386 expression data points per participant, with each point consisting of a gene expression level value for a given gene. Seven hundred and forty-four participants with corresponding diagnosis and expression data were utilized. The goal of the implementation was to separate gene expressions by diagnosis (NC, MCI, and AD), build a network for each condition, and compare networks. With 49,386 samples per participant, the full set of correlation values became extremely large, so the data needed to be filtered. This was done using Bioconductor’s genefilter R utility (http://www.bioconductor.org/packages/release/bioc/html/genefilter.html). Genes were kept if they had both a coefficient of variation between 0.7 and 10, and if 20% or more samples exhibited an expression level greater than 100 for that gene [45]. Because data was normalized using the RMA (Robust Multi-chip Average) normalization method, the normalization and the log-scaling was reversed for the calculations. Bioconductor’s genefilter was also used to run an ANOVA between groups and apply a filter, keeping genes with the most significant differences between conditions. Fifty genes passed both filtering stages. A list of these genes is available in Table 1, along with their overall expression change in AD and MCI participants relative to NC participants. *p* values for these expression changes are listed if they exceeded 0.1. Three Pearson Correlation matrices were generated for both positive and negative correlations between genes of the different conditions (NC, MCI, and AD), resulting in six total matrices. Thresholds were applied to the correlation matrices in order to filter out the very weak relationships. For the purpose of analyzing relationship-disruptions between disease states, a threshold of 0.1 was applied to filter out very weak or non-correlations. For the purpose of directly examining the relationships between genes, a threshold of 0.3 was applied to filter out the weak correlations.

Networks were then generated in Cytoscape [46]. In order to better understand these networks, two complementary visual analysis tools were used. DyNet is a tool that compares two or more networks with the same node-set and identifies the nodes whose connections change the most between the different networks (rewired) The higher the rewiring score, the more the gene’s co-expression with other genes varies between conditions. Diffany is another tool that was used in order to compare networks that functions differently than DyNet [47]. Diffany was used to generate a directional network that visualizes how the correlations between genes changes between the normal and afflicted (MCI and AD) states. The steps from data acquisition to network creation are detailed in Figure 5 below.

## 5. Conclusions

Conducting a network-based analysis of the gene expression levels and the co-expression patterns observed between samples in blood samples obtained from NC, MCI, and AD subjects has proven to be fairly productive. In addition to reinforcing some of the research performed on many of the genes already, this work and other network-based analysis serve to elucidate some other potential genes and pathways for further study; many of the connections are not obvious at a glance. All of the genes examined in this study exhibited aberrant expression levels in those with Alzheimer’s disease and mild cognitive impairment. While some may have not been connected on the networks examined in this study, this does not necessarily mean it is insignificant nor that it exhibits no patterns of co-expression. They may be related to other genes that were excluded in this study, due to the filtration applied or limitations of the microarray analysis. Additionally, this study examined genes found in the subject’s blood, which may not translate directly to expression within the subject’s brain in all cases. Many of the known key players in Alzheimer’s disease, such as APP, are differentially expressed between tissue types and between different regions of the brain [48]. With that being said, with further study these genes may prove to be valuable biomarkers for use in much-needed early diagnosis tests, as many of the genes show similar expression patterns and relationships in both MCI and AD individuals and blood is a relatively easily obtained sample.

## Figures and Tables

**Figure 1 ijms-21-00332-f001:**
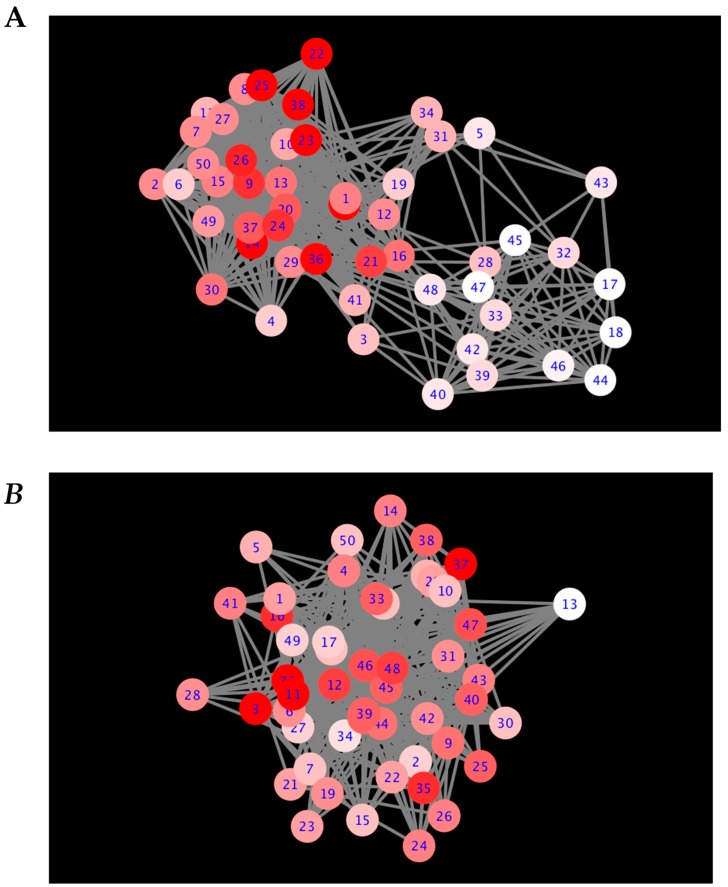
DyNet networks. The deeper the red of the node, the more rewiring has occurred. DyNet calculates the variance between each node’s connectivity between networks and computes a score based on the number of altered (i.e., added, removed) connections. Based on Pearson coefficient threshold *T* = 0.1 networks. (**A**) Positive Pearson correlation DyNet network. (**B**) Negative Pearson correlation DyNet network.

**Figure 2 ijms-21-00332-f002:**
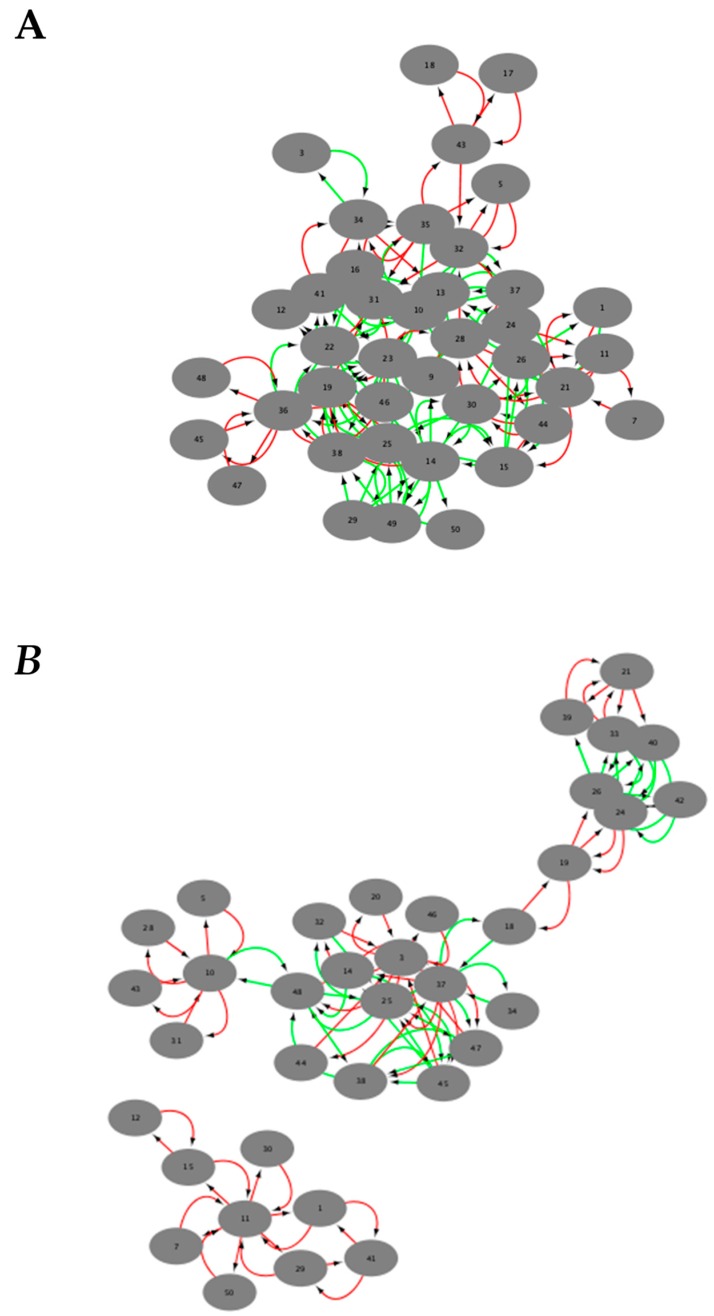
Diffany networks. Green arrows represent increase in association and red indicate decrease in association between genes (Alzheimer’s disease (AD)/MCI vs. NC). Association is determined by the addition or removal of edges between networks in comparison to a reference condition. Based on Pearson coefficient threshold *T* = 0.1 networks. (**A**) Diffany network generated from positive correlation networks. (**B**) Diffany network generated from negative correlation networks.

**Figure 3 ijms-21-00332-f003:**
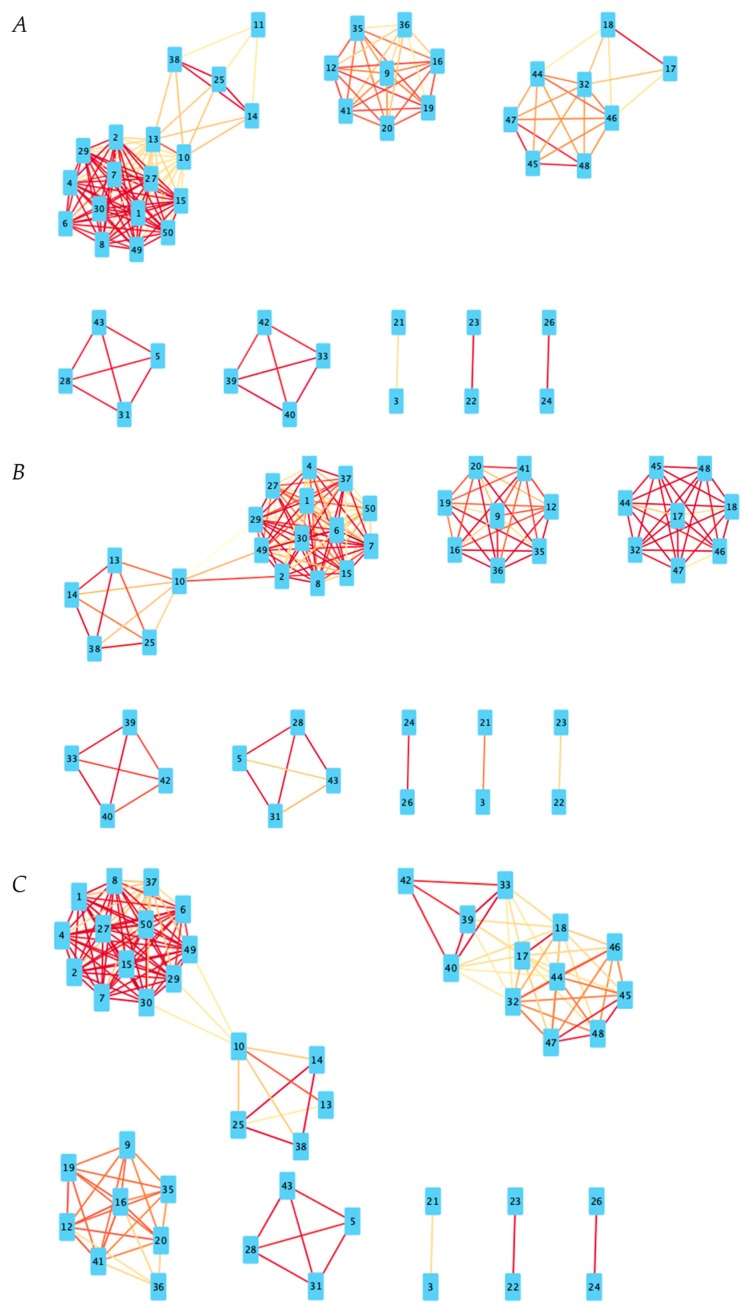
Positive correlation networks for all disease states. Red lines indicate a higher Pearson Correlation coefficient. Based on Pearson coefficient threshold *T* = 0.3 networks. (**A**) Positive correlation network for NC state. (**B**) Positive correlation network for MCI state. (**C**) Positive correlation network for AD state.

**Figure 4 ijms-21-00332-f004:**
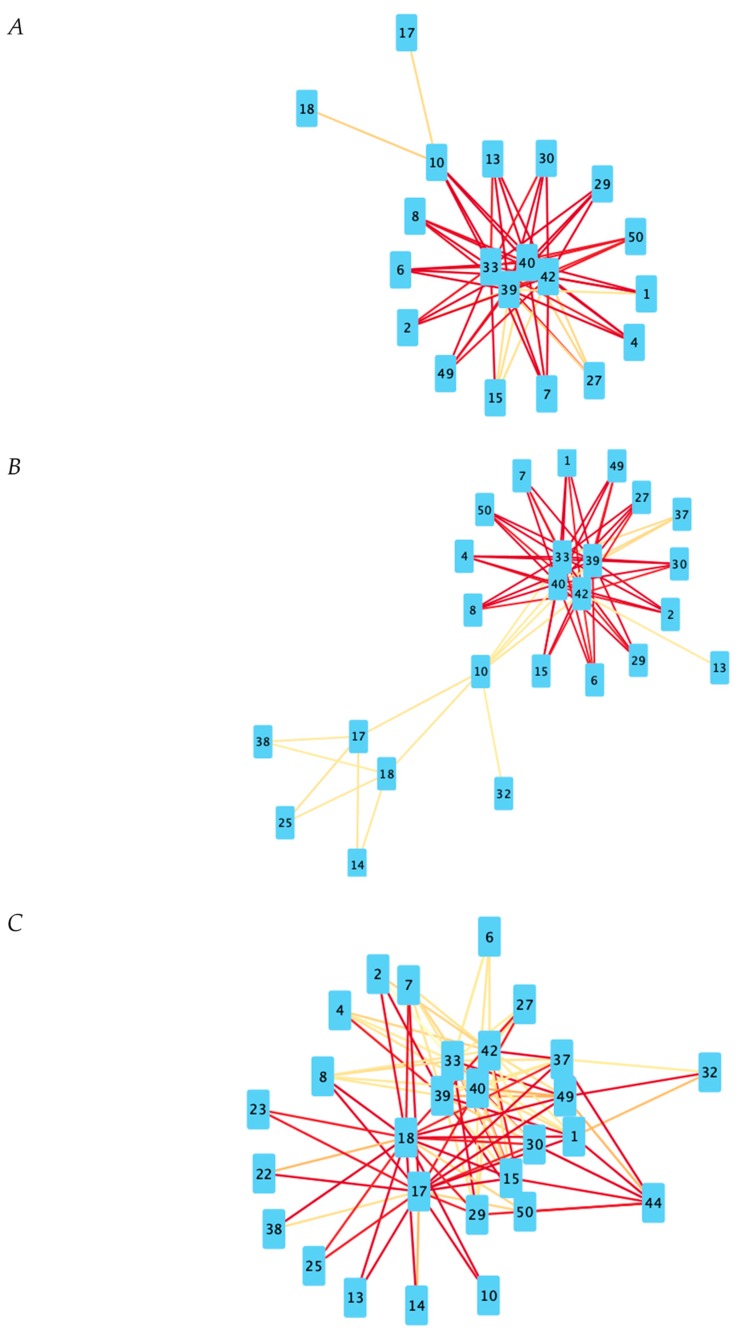
Negative correlation networks for all disease states. Red lines indicate a higher Pearson correlation coefficient. Based on Pearson coefficient threshold *T* = 0.3 networks. (**A**) Negative correlation network for NC state. (**B**) Negative correlation network for MCI state. (**C**) Negative correlation network for AD state.

**Figure 5 ijms-21-00332-f005:**
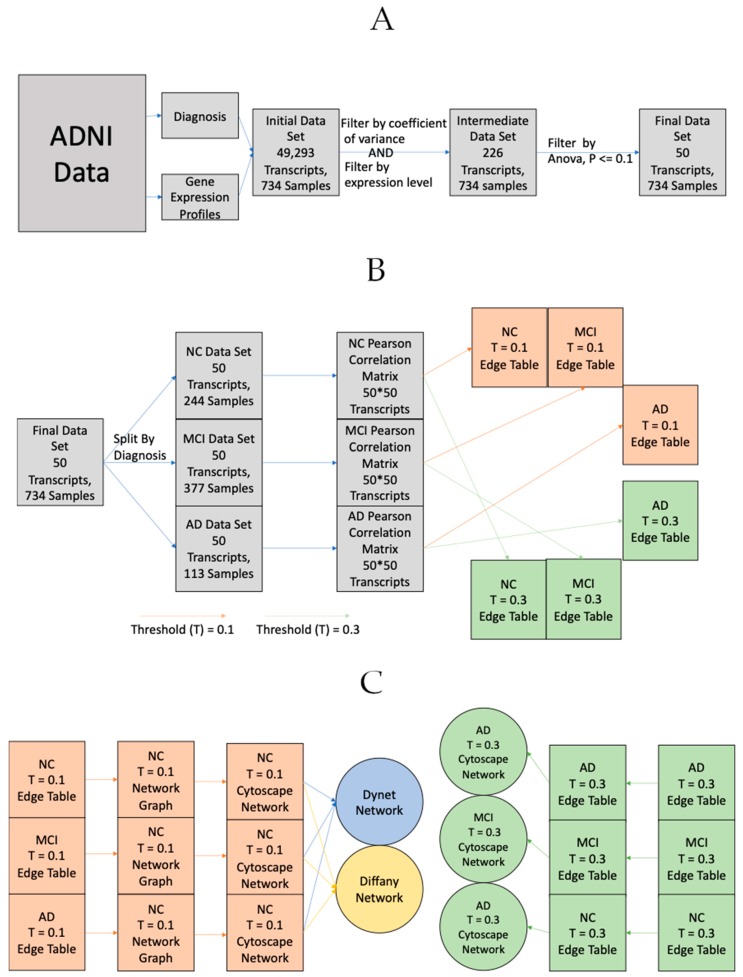
The general process for retrieving, processing and ultimately turning Alzheimer’s Disease Neuroimaging Initiative (ADNI) gene expression data into their corresponding networks. (**A**) General process for acquisition/processing and gene filtration. (**B**) General process for transforming expression data to edge tables. The process is the same for both positive and negative correlation matricies/networks, only the sign and direction of the threshold is changed. e.g., >0.3 for positive correlation, <−0.3 for negative correlation. (**C**) General process for transforming gene expression data from edge tables to networks. The process is the same for both positive and negative correlation matricies/networks, only the sign of the threshold is changed. e.g., >0.3 for positive correlation, <−0.3 for negative correlation. *T* = 0.1 edge tables were used for the DyNet (Figure 1) and Diffany (Figure 2) networks, *T* = 0.3 edge tables were used for the basic networks (Figure 3 and Figure 4).

**Table 1 ijms-21-00332-t001:** Genes selected using genefilter | Expression Rel. to NC ^1^.

Key	Gene	Expression in MCI ^2^	Expression in AD
1	[*RPS4Y1*] Ribosomal protein S4 Y-linked 1 Probe set: 11716411_x_at	Up	Up
2	[*EIF1AY*] eukaryotic translation initiation factor 1A Y-linked Probe set: 11720807_x_at	Up	Up
3	[*GATA2*] GATA binding protein 2 Probe set: 11722761_a_at	Down	Down *p* = 0.203
4	[*DDX3Y*] DEAD (Asp-Glu-Ala-Asp) box helicase 3 Y-linked Probe set: 11724075_a_at	Up	Up
5	[*HLA-DQA1*] Major histocompatibility complex class II DQ alpha 1 Probe set: 11724799_x_at	Up	Up *p* = 0.694
6	[*USP9Y*] ubiquitin specific peptidase 9 Y-linked Probe set: 11725294_at	Up	Up *p* = 0.194
7	[*KDM5D*] lysine (K)-specific demethylase 5D Probe set: 11726813_a_at	Up	Up
8	[*KDM5D*] lysine (K)-specific demethylase 5D Probe set: 11726814_x_at	Up	Up
9	[*TBC1D22B*] TBC1 domain family member 22B Probe set: 11728078_a_at	Up *p* = 0.670	Down
10	[*BPI*] Bactericidal/permeability-increasing protein Probe set: 11729344_at	Up *p* = 0.225	Up
11	[*ANKRD22*] Ankyrin repeat domain 22 Probe set: 11732425_at	Up *p* = 0.334	Up
12	[*TMOD1*] Tropomodulin 1 Probe set: 11732501_a_at	Down *p* = 0.587	Down
13	[*DEFA4*] Defensin alpha 4 corticostatin Probe set:11732546_at	Up *p* = 0.293	Up
14	[*HP*] Haptoglobin Probe set: 11733829_x_at	- *p* = 0.991	Up
15	[*DDX3Y*] DEAD (Asp-Glu-Ala-Asp) box helicase 3 Y-linked Probe set: 11734664_x_at	Up	Up
16	[*OSBP2*] Oxysterol binding protein 2 Probe set: 11736205_a_at	Down *p* = 0.473	Down
17	[*FCRL1*] Fc receptor-like 1 Probe set: 11736882_a_at	Down *p* = 0.641	Down
18	[*FCRL1*] Fc receptor-like 1 Probe set: 11736883_x_at	Down *p* = 0.694	Down
19	[*FAM46C*] Family with sequence similarity 46 member C Probe set: 11739338_at	Down *p* = 0.531	Down
20	[*OR2W3*] (Locus via Non-standard RNA) olfactory receptor family 2 subfamily W member 3 Probe set: 11741636_at	- *p* = 0.843	Down
21	[*CACNG6*] Calcium channel voltage-dependent gamma subunit 6 Probe set: 11742124_a_atex	Down	Down *p* = 0.616
22	[*FOLR3*] folate receptor 3 (gamma) Probe set: 11744140_a_at	Up *p* = 0.645	Up *p* = 0.163
23	[*FOLR3*] folate receptor 3 (gamma) Probe set: 11744141_x_at	Up *p* = 0.582	Up *p* = 0.137
24	[*CBS*] Cystathionine-beta-synthase Probe set: 11744286_s_at	- *p* = 0.995	Up
25	[*HP*] Haptoglobin Probe set: 11744649_x_at	- *p* = 0.76	Up *p* = 0.141
26	[*CBS*] Cystathionine-beta-synthase Probe set: 11744835_s_at	- *p* = 0.995	Up
27	[*KDM5D*] lysine (K)-specific demethylase 5D Probe set: 11745012_a_at	Up	Up
28	[*HLA-DQB1*] Major histocompatibility complex class II DQ beta 1 Probe set: 11746804_x_at	Up	Up *p* = 0.46
29	[*DDX3Y*] DEAD (Asp-Glu-Ala-Asp) box helicase 3 Y-linked Probe set: 11748424_x_at	Up	Up
30	[*DDX3Y*] DEAD (Asp-Glu-Ala-Asp) box helicase 3 Y-linked Probe set: 11749841_x_at	Up	Up
31	[*HLA-DQA1*] Major histocompatibility complex class II DQ alpha 1 Probe set: 11750528_x_at	Up	- *p* = 0.89
32	[ENSG00000211625 || ENSG00000239951] (Matches 2 Loci; Matches Ensembl Gene) Putative uncharacterized protein ENSP00000374805 [Source:UniProtKB/TrEMBL;Acc:A6NLY3] || Ig kappa chain V-III region HAH Precursor [Source:UniProtKB/Swiss-Prot;Acc:P18135] Probe set: 11753832_x_at	Down *p* = 0.123	Down
33	[*XIST*] X inactive specific transcript (non-protein coding) Probe set: 11754194_s_at	Down	Down
34	[*EGR1*] Early growth response 1 Probe set: 11754334_s_at	Down	Down
35	[*NUDT4* || *NUDT4P2* || *NUDT4P1*] (Matches 3 Loci) Nudix (Nucleoside diphosphate linked moiety X)-type motif 4 || nudix (nucleoside diphosphate linked moiety X)-type motif 4 pseudogene 2 || Nudix (nucleoside diphosphate linked moiety X)-type motif 4 pseudogene 1 Probe set: 11754453_s_at	- *p* = 0.846	Down
36	[*SHISA4*] shisa family member 4 Probe set: 11756240_a_at	Down *p* = 0.641	Down
37	[*PTGDS*] prostaglandin D2 synthase 21kDa (brain) Probe set: 11756587_a_at	- *p* = 0.853	Up
38	[*HP*] Haptoglobin Probe set: 11757277_x_at	- *p* = 0.932	Up
39	[*XIST*] X inactive specific transcript (non-protein coding) Probe set: 11757733_s_at	Down	Down
40	[*XIST*] X inactive specific transcript (non-protein coding) Probe set: 11757857_s_at	Down	Down
41	[*TRIM10*] tripartite motif containing 10 Probe set: 11758611_s_at	Up *p* = 0.647	Down
42	(Matches Non-standard RNA) JARID1C protein (JARID1C) mRNA complete cds alternatively spliced Probe set: 11761133_at	Down	Down
43	[*HLA-DQB1*] (POOR HIT 44%) Major histocompatibility complex class II DQ beta 1 Probe set: 11762641_x_at	Up	Up *p* = 0.56
44	(DEPRECATED TARGET; Matches RefSeq) (Deprecated) PREDICTED: Homo sapiens similar to *hCG2042707* (LOC650405) || (Deprecated) PREDICTED: Homo sapiens similar to pre-B lymphocyte gene 1 (LOC652493) || (Deprecated) PREDICTED: Homo sapiens similar to *hCG26659* (LOC100291464) Probe set: 11763222_x_at	Down *p* = 0.316	Down
45	[*ENSG00000211663*] (Matches Ensembl Gene) V2-13 protein Fragment [Source:UniProtKB/TrEMBL; Acc:Q5NV73] Probe set: 11763229_x_at	Down *p* = 0.331	Down
46	[*ENSG00000242534* || *ENSG00000244116*] (Matches 2 Loci; Matches Ensembl Gene) immunoglobulin kappa variable 2D-28 [ENST00000453166 ENST00000558026] || immunoglobulin kappa variable 2-28 [ENST00000482769]	Up *p* = 0.614	Down *p* = 0.122
47	[*ENSG00000211663*] (Matches Ensembl Gene) V2-13 protein Fragment [Source:UniProtKB/TrEMBL;Acc:Q5NV73] Probe set: 11763255_x_at	Down *p* = 0.279	Down
48	[*ENSG00000211663*] (Matches Ensembl Gene) V2-13 protein Fragment [Source:UniProtKB/TrEMBL;Acc:Q5NV73] Probe set: 11763551_x_at	Down *p* = 0.332	Down
49	(Matches Non-standard RNA) mRNA; cDNA DKFZp686L12190 (from clone DKFZp686L12190): Probe set: 11763837_s_at	Up	Up
50	[*TXLNG2P*] (Matches Ensembl Gene) Uncharacterized protein CYorf15B (Lipopolysaccaride-specific response 5-like protein) [Source:UniProtKB/Swiss-Prot;Acc:Q9BZA4] Probe set: 11764064_s_at	Up	Up

^1^ NC—normal condition. ^2^ MCI—mild cognitive impairment.

**Table 2 ijms-21-00332-t002:** DyNet top positive rewiring genes.

Gene	DyNet Rewiring Score
[*HP*] Haptoglobin Probe set: 11757277_x_at	8.33
[*FOLR3*] folate receptor 3 (gamma) Probe set: 11744141_x_at	8.00
[*FOLR3*] folate receptor 3 (gamma) Probe set: 11744140_a_at	8.00
[*NUDT4* || *NUDT4P2* || *NUDT4P1*] (Matches 3 Loci) Nudix (Nucleoside diphosphate linked moiety X)-type motif 4 || nudix (nucleoside diphosphate linked moiety X)-type motif 4 pseudogene 2 || Nudix (nucleoside diphosphate linked moiety X)-type motif 4 pseudogene 1HP Probe set: 11754453_s_at	7.00
[*HP*] Haptoglobin Probe set: 11744649_x_at	7.00
[*HP*] Haptoglobin Probe set: 11733829_x_at	7.00
[*SHISA4*] shisa family member 4 Probe set: 11756240_a_at	6.67
[*CBS*] Cystathionine-beta-synthase Probe set: 11744835_s_at	6.00
[*TBC1D22B*] TBC1 domain family member 22B Probe set: 11728078_a_at	5.67
[*CBS*] Cystathionine-beta-synthase Probe set: 11744286_s_at	5.67

**Table 3 ijms-21-00332-t003:** DyNet top negative rewiring genes.

Gene	DyNet Rewiring Score
[*GATA2*] GATA binding protein 2 Probe set: 11722761_a_at	8.00
[*PTGDS*] prostaglandin D2 synthase 21kDa (brain) Probe set: 11756587_a_at	6.33
[*ANKRD22*] Ankyrin repeat domain 22 Probe set: 11732425_at	6.33
[*SHISA4*] shisa family member 4 Probe set: 11756240_a_at	5.67

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
