# Peer review of "Network Medicine Approach for Analysis of Alzheimer’s Disease Gene Expression Data"

_ijms, 2020, doi:10.3390/ijms21010332_

Round 1

Reviewer 1 Report

This is a very interesting manuscript and describes the use of a network medicine approach to identifying potentially important changes to gene expression in mild cognitive impairment and Alzheimer's patients through analysis of blood. They uncovered some interesting new insights and also generated further support for known pathways, thus validating the outcomes. I believe that overall this is an important and interesting paper, however, there are a few areas that need some more clarification to enable the study to be more easily interpreted by the reader.

In table 1, there are a number of duplicate proteins that come up e.g. HP, but there is no explanation as to what this means, and how you can have the same gene come up again. A clear explanation of this would be helpful. It seems odd and perhaps counter-intuitive that there is actually not a lot of difference between MCI and AD, with most genes changing the same. Are there differences in the level of those changes? Given that the disease has obviously progressed considerably in AD compared to MCI, can the authors explain why there are not more identified differences? In relation to the above comment, I feel that the study would be greatly strengthened if the authors are able to generate a data set for an unrelated neurological disorder? This would provide more information on the specificity of the changes for MCI/AD rather than just neurological or other disease. It would also help with clarity if the authors provided a deeper explanation of the figures before talking about the data in the table. It takes quite an effort to work out how the data in the figures lead to the outcomes in the table. As above, a clear explanation in the results of what a DyNet rewiring score means would help with overall interpretation for a broad audience.

Minor: in the abstract the authors say that Alzheimer's disease is 'often' fatal. This should be changed to 'always' fatal.

Author Response

Reviewer 1

Many thanks for your critiques and suggestions. Here are our point-by-point responses:

“In table 1, there are a number of duplicate proteins that come up e.g. HP, but there is no explanation as to what this means, and how you can have the same gene come up again. A clear explanation of this would be helpful.”

Response: The duplicate genes come from different probe sets. We have more thoroughly explained this in the text and adjusted the tables accordingly.

“It seems odd and perhaps counter-intuitive that there is actually not a lot of difference between MCI and AD, with most genes changing the same. Are there differences in the level of those changes? Given that the disease has obviously progressed considerably in AD compared to MCI, can the authors explain why there are not more identified differences? In relation to the above comment, I feel that the study would be greatly strengthened if the authors are able to generate a data set for an unrelated neurological disorder? This would provide more information on the specificity of the changes for MCI/AD rather than just neurological or other disease.”

Response: While having gene expression data for another disease would be ideal, acquiring a set with the same transcripts as the ADNI study would be difficult given time constraints. However, we believe that including the direction of expression changes in AD and MCI (table 1) without other descriptors was inadequate. As such, we have added the adjusted P-values for the expression differences to the table where it was greater than 0.1. This should make it much clearer that, while there are certainly commonalities between the MCI and AD states, they are not as similar as the original table made it seem. We apologize for the confusion.

“It would also help with clarity if the authors provided a deeper explanation of the figures before talking about the data in the table. It takes quite an effort to work out how the data in the figures lead to the outcomes in the table. As above, a clear explanation in the results of what a DyNet rewiring score means would help with overall interpretation for a broad audience.”

Response: We have more thoroughly explained the different networks, and how to interpret them (I.e., what the color of the node means, what the presence of an edge in the Diffany and DyNet networks means, etc.) This should make it much clearer what their significance is and how the results were drawn from them.

“Minor: in the abstract the authors say that Alzheimer's disease is 'often' fatal. This should be changed to 'always' fatal.”

Response: We have fixed this error and a similar issue in the abstract; thank you for pointing this out.

Reviewer 2 Report

The authors analyze gene expression data from Alzheimer’s Disease Neuroimaging Initiative (ADNI)

Major remarks

1. “we have applied the cutting-edge technology of network medicine”

I feel generating coexpression networks based on simple correlations between genes using two previously developed tools (i.e., DyNet and Diffany among many others) does not fulfill the promise of conducting cutting-edge network medicine analysis. The manuscript builds up on rather trivial differential expression & co-expression analyses and does not the presented findings in the discussion lack statistical evidence in regards to their soundness, making the entire text a largely speculative piece.

2. DyNet and Diffany are only introduced later in section 4.2 and referred without any previous explanation in the results (section 2). Please briefly introduce them in results. Similarly state the correspondence to mild cognitive impairment (MCI) and normal condition (NC) earlier. It is also not clear what exactly the two methods do and what are the scores they produce correspond to.

3. “Both tables were cut off at the score of 5. Some genes with scores lower than five were considered for further analysis; those with scores nearing five were takin into consideration, one such gene being OSBP2.” What does this score refer to? Why the cutoff of 5 is chosen?

4. “Thresholds were applied to the correlation matrices in order to filter out the very weak relationships. For the purpose of analyzing relationship-disruptions between disease states, a threshold of 0.1 was applied. For the purpose of directly examining the relationships between genes, a threshold of 0.3 was applied to filter out the weak correlations.”

Justify the chosen arbitrary cutoffs

Needs clarification and/or language revision

- with 64 removed due to QC failure

- In order to properly utilize these data, diagnosis was used from the date of gene expression collection and the expression dataset required cleaning

- This process of filtering left 50 genes remaining

- gene expression levels observed in blood samples obtained from NC, MCI and AD subjects has proven to be fairly productive

- it serves to elucidate some other potential genes and pathways for further study; many of the connections are not obvious at a glance

- so while some may have not been connected on the networks examined, they may still be related in some way through other genes that were excluded from this study

- nearing five were takin** into consideration

Author Response

Reviewer 2

Many thanks for your critiques and suggestions. Here are our point-by-point responses:

The authors analyze gene expression data from Alzheimer’s Disease Neuroimaging Initiative (ADNI)

Major remarks

“1. “we have applied the cutting-edge technology of network medicine”

I feel generating coexpression networks based on simple correlations between genes using two previously developed tools (i.e., DyNet and Diffany among many others) does not fulfill the promise of conducting cutting-edge network medicine analysis. The manuscript builds up on rather trivial differential expression & co-expression analyses and does not the presented findings in the discussion lack statistical evidence in regards to their soundness, making the entire text a largely speculative piece.”

Response: We have introduced new, more thorough explanations of the networks used to draw our conclusions. We hope that this makes our overall results/discussion more sensible. However, the abstract has been changed to take a more modest stance on our application of networks to gene expression data.

“2. DyNet and Diffany are only introduced later in section 4.2 and referred without any previous explanation in the results (section 2). Please briefly introduce them in results. Similarly state the correspondence to mild cognitive impairment (MCI) and normal condition (NC) earlier. It is also not clear what exactly the two methods do and what are the scores they produce correspond to.”

Response: We have more thoroughly explained (with examples) what the meaning of the DyNet and Diffany networks (what node color means, what the presence of an edge means, rewiring score, etc.) in the text.

“3. “Both tables were cut off at the score of 5. Some genes with scores lower than five were considered for further analysis; those with scores nearing five were takin into consideration, one such gene being OSBP2.” What does this score refer to? Why the cutoff of 5 is chosen?”

Response: The rewiring score, as well as the purpose + methods of the DyNet networks in general, now have more comprehensive explanations in the text. The cutoff of 5 was chosen to limit the number of genes in the table to 10.

“4. “Thresholds were applied to the correlation matrices in order to filter out the very weak relationships. For the purpose of analyzing relationship-disruptions between disease states, a threshold of 0.1 was applied. For the purpose of directly examining the relationships between genes, a threshold of 0.3 was applied to filter out the weak correlations.”

Justify the chosen arbitrary cutoffs”

Response: These cutoffs were chosen as correlation coefficients below 0.1 are generally considered to represent either very weak or non-existent correlations. 0.3 is generally the lower bound for moderate-strong correlations.

Needs clarification and/or language revision

- with 64 removed due to QC failure

Changed wording to make the statement more clear

- In order to properly utilize these data, diagnosis was used from the date of gene expression collection and the expression dataset required cleaning

Changed wording to make the statement more clear

- This process of filtering left 50 genes remaining

Changed wording to make the statement more clear

- gene expression levels observed in blood samples obtained from NC, MCI and AD subjects has proven to be fairly productive

Changed wording to make the statement more clear

- it serves to elucidate some other potential genes and pathways for further study; many of the connections are not obvious at a glance

Changed wording to make the statement more clear

- so while some may have not been connected on the networks examined, they may still be related in some way through other genes that were excluded from this study

Changed wording to make the statement more clear

- nearing five were takin** into consideration

Spelling error corrected

Round 2

Reviewer 2 Report

I thank the authors for taking into account my earlier comments to the manuscript. Despite the revisions, I still feel that the article still lacks the information on the choice & optimization of the thresholds  (see my earlier comments 3&4) and how they would affect the analysis and the conclusions derived from it. I also do not follow the data presented on Table 1 as it is not clear how these genes are identified. The added text does not help either:

"All numbers correspond to genes in table 1 by key. Genes that are
repeated in the dataset represent different probe sets, which are specified in the table. P values greater than 0.1 are listed below the direction of relative expression. A “-“ indicates the P value exceeded 0.75, regardless of direction (Welch t-test, FDR adjusted P value)."

From this text, it sounds genes with a "P > 0.1" is selected, which is the opposite definition of significance (and it is not clear whether the values on the table are adjusted P values). Which contradicts with the following text:

"Looking at table 1, we can see that in many of the genes that are significantly upregulated and downregulated in AD individuals are regulated in the same direction in MCI individuals, though there are some notable exceptions such as haptoglobin, along with genes whose expression changes are not necessarily significant."

I find the following statement somewhat misleading given that there are many network-based studies investigating Alzheimer's disease pathology in the recent literature:

"While studies involving gene expression data from AD patients are not rare, there are comparatively few that conduct network-based analysis on that data."

I highlight several relevant studies (few selected from the last 5 years), IMHO they should be included either in the introduction or in the discussion to highlight the difference between them and the presented work.

Siavelis et al 2016 Brief Bics (Review article) doi: 10.1093/bib/bbv048

Kitsak et al. 2016 Sci Rep doi: 10.1038/srep35241

Mostafavi et al 2018 Nat Neuro doi: 10.1038/s41593-018-0154-9

Hosp et al 2015 Cell Rep doi: 10.1016/j.celrep.2015.04.030

Aguirre-Plans et al 2019 Pharmaceu doi: 10.3390/ph11030061

Author Response

Reviewer 2

Thank you again for your critiques and sugestions.

“I thank the authors for taking into account my earlier comments to the manuscript. Despite the revisions, I still feel that the article still lacks the information on the choice & optimization of the thresholds (see my earlier comments 3&4) and how they would affect the analysis and the conclusions derived from it.”

Response: The correlation coefficient threshold of 0.1 was used was used for the creation of the Diffany and DyNet as that is generally considered to be the boundary for irrelevant correlations (see this recent review [1]) As the bulk of the analysis of the DyNet networks came from the nodes themselves, they can be interpreted visually by the reader easily, even with a large number of edges. The Diffany network is also clear enough to be interpreted visually by the reader. The threshold was not raised so the respective algorithms of Diffany and DyNet would still take them into account.

This threshold was raised for the simple co-expression networks in figures 3 and 4 as much of the analysis was focused on the stronger correlations, and keeping the threshold at 0.1 would add considerable visual noise. 0.3 was chosen to eliminate edges for weak correlations. Unfortunately, there is no consensus boundary for weak/moderate correlations [1], so one had to be chosen somewhat arbitrarily. We chose to err on the side of including more edges rather than excluding them.

The thresholds for table 2 in the paper were not used in the analysis but were for keeping the table size to a reasonable level (10 transcripts maximum). This was done as the Dynet networks are fairly easy for the reader to analyze visually, with the table giving an idea of the “scale” for the color gradient.

“I also do not follow the data presented on Table 1 as it is not clear how these genes are identified.”

Response: Genes are identified by both their name and the ID of the probe set used. Originally we had listed the probe set ID only if there was a duplicate in the set, we have changed this so all IDs are listed.

We have added a brief description of the gene selection/filtering process explained in the methods section immediately prior to the table. Additionally, some changes have been made to non-specific wording that may have added to the confusion.

“I find the following statement somewhat misleading given that there are many network-based studies investigating Alzheimer's disease pathology in the recent literature: I highlight several relevant studies (few selected from the last 5 years), IMHO they should be included either in the introduction or in the discussion to highlight the difference between them and the presented work.

Siavelis et al 2016 Brief Bics (Review article) doi: 10.1093/bib/bbv048

Kitsak et al. 2016 Sci Rep doi: 10.1038/srep35241

Mostafavi et al 2018 Nat Neuro doi: 10.1038/s41593-018-0154-9

Hosp et al 2015 Cell Rep doi: 10.1016/j.celrep.2015.04.030

Aguirre-Plans et al 2019 Pharmaceu doi: 10.3390/ph11030061”

Response: We have taken the references you have generously provided and included a brief summary of prior work, as suggested. Agguire-Plans et. al. was not specifically focused on AD, and so was excluded from this summary, but we have referenced the work’s relevant information in the conclusion.

Schober, P.; Boer, C.; Schwarte, L.A. Correlation Coefficients: Appropriate Use and Interpretation. Anesthesia & Analgesia 2018, 126, 1763-1768, doi:10.1213/ane.0000000000002864.

Round 3

Reviewer 2 Report

I thank the authors for their response to the points I had raised for the eariler version of the manuscript. I feel the manuscript meets expected standards of the journal and could bring network-centric insights to the pathology of AD.